# Meiotic DNA exchanges in *C. elegans* are promoted by proximity to the synaptonemal complex

David E Almanzar, Spencer G Gordon*, Chloe Bristow*, Antonia Hamrick*, Lexy von Diezmann, Hanwenheng Liu [iD],
Ofer Rog [iD]

**During meiosis, programmed double-strand DNA breaks are repaired to form exchanges between the parental chromosomes called crossovers. Chromosomes lacking a crossover fail to segregate accurately into the gametes, leading to aneuploidy. In addition to engaging the homolog, crossover formation requires the promotion of exchanges, rather than non-exchanges, as repair products. However, the mechanism underlying this meiosis-specific preference is not fully understood. Here, we study the regulation of meiotic sister chromatid exchanges in *Caenorhabditis elegans* by direct visualization. We find that a conserved chromosomal interface that promotes exchanges between the parental chromosomes, the synaptonemal complex, can also promote exchanges between the sister chromatids. In both cases, exchanges depend on the recruitment of the same set of pro-exchange factors to repair sites. Surprisingly, although the synaptonemal complex usually assembles between the two DNA molecules undergoing an exchange, its activity does not rely on a specific chromosome conformation. This suggests that the synaptonemal complex regulates exchanges—both crossovers and sister exchanges—by establishing a nuclear domain conducive to nearby recruitment of exchange-promoting factors.**

## Introduction

Meiosis is the specialized cell division that creates haploid gametes from diploid precursor cells. For the paternal and maternal copies of each chromosome (homologs) to segregate accurately, they must form at least one reciprocal inter-homolog exchange (crossover). At the same time, the complex DNA acrobatics that generate crossovers carry a risk of corrupting the genome. As a result, crossover formation is tightly regulated.

Crossovers form when programmed DNA double-strand breaks (DSBs) are mended by a conserved repair pathway called homologous recombination, which restores genomic integrity using information from a template that shares sequence homology. Although homologous recombination throughout development uses many of the same molecular actors and DNA intermediates, their regulation during meiosis is distinct in two respects. The first is the choice of repair template: although mitotic cells use the identical copy available in the sister chromatid, meiotic cells preferentially use the similar homolog as a repair template (Humphryes & Hochwagen, 2014). The second is the nature of the repair products. Repair events can be resolved to reciprocally exchange flanking sequences or as non-exchange products that locally patch the DSB. Exchanges jeopardize both the broken DNA molecule and the template and are indeed avoided in mitotic cells, where they are associated with cancer predisposition (Chaganti et al, 1974). In meiosis, however, a tightly regulated number of inter-homolog repair intermediates (in *C. elegans*, exactly one per chromosome) are processed as exchanges to form crossovers, whereas most other repair events are resolved as non-exchanges (Pyatnitskaya et al, 2019).

The mechanism that promotes the formation of a precise number of crossovers while channeling other repair events into a non-exchange fate depends on a set of pro-crossover factors known as the ZMMs (named after the budding yeast proteins Zip1-4, Mer3, Msh4-5, and Spo16; the *C. elegans* components, which we also refer to as ZMMs, are ZHP-1-4, COSA-1, and MSH-5-HIM-14[MSH-4] [Zalevsky et al, 1999; Kelly et al, 2000; Börner et al, 2004; Jantsch et al, 2004; Yokoo et al, 2012; Zhang et al, 2018; Pyatnitskaya et al, 2019]). Although ZMMs in some lineages play a role in the initial pairing of the homologs (Pyatnitskaya et al, 2019), their seemingly conserved function is regulating repair outcomes. ZMMs form foci at a subset of inter-homolog repair sites—called recombination nodules—that correspond in their number and distribution to crossovers (Carpenter, 1975; Zickler & Kleckner, 1999; Börner et al, 2004; Yokoo et al, 2012). Recombination nodules designate crossovers by regulating the processing of repair intermediates localized within them to yield exchanges. Indeed, depending on the lineage, removing the ZMMs results in a drastically altered number and distribution of crossovers or their absence altogether (Pyatnitskaya et al, 2019).

School of Biological Sciences and Center for Cell and Genome Sciences, University of Utah, Salt Lake City, UT, USA

Correspondence: ofer.rog@utah.edu
David E Almanzar's present address is Myriad Genetics, Salt Lake City, UT, USA
Hanwenheng Liu's present address is The Division of Biology and Biomedical Sciences, Washington University in St. Louis, St. Louis, MO, USA
*Spencer G Gordon, Chloe Bristow, and Antonia Hamrick contributed equally to this work

Key to the regulation of recombination nodules is a conserved chromosomal interface—the synaptonemal complex—that localizes between the homologs. Once the homologs independently assemble their chromatin around a backbone called the axis and find each other, the central region of the synaptonemal complex (referred to throughout as SC-CR) aligns them and assembles along their lengths (MacQueen et al, 2002; Rog & Dernburg, 2013). Although the SC-CR plays a conserved role in regulating crossovers, the mechanisms by which the SC-CR controls recombination nodules seem to vary in different lineages. In worms, both the SC-CR and DSBs are necessary for the formation of recombination nodules: when a chromosome lacks DSBs or fails to assemble the SC-CR, it does not form recombination nodules and does not undergo crossovers (Yokoo et al, 2012; Rosu et al, 2013; Stamper et al, 2013). However, conditions that eliminate DSBs or the SC-CR throughout the nucleus reveal that each contributes independently to recombination nodule formation. In animals that lack the SC-CR, ZMM proteins form DSB-dependent foci on chromosomes ([Li et al, 2018; Cahoon et al, 2019]; similar findings have been made in plants [Durand et al, 2022] and yeast [Fung et al, 2004]), and in the absence of DSBs, ZMM proteins form a focus abutting chromatin-free aggregates of SC-CR material ([Rog et al, 2017; Zhang et al, 2018]; similar foci have been documented in yeast [Tsubouchi et al, 2006; Shinohara et al, 2015]). However, the molecular mechanisms by which the SC-CR and DSBs control crossover formation remain poorly understood, in worms and other organisms.

Recently, we showed that exchanges between sister chromatids are rare during meiosis in *C. elegans* (Almanzar et al, 2021). Interestingly, we found that the SC-CR can promote sister exchanges when it mislocalizes to unpaired chromosomes (i.e., chromosomes that cannot undergo crossovers [Cahoon et al, 2019; Almanzar et al, 2021]). Sister exchanges were elevated concomitantly with the formation of recombination nodules, raising the possibility that recombination nodules can promote exchanges between sisters.

Here, we study the mechanism by which the SC-CR promotes sister exchanges in *C. elegans*. By testing several conditions where the SC-CR assembles on chromosomes that exclusively undergo sister-directed repair, we found elevated rates of sister exchanges that correspond to the number of recombination nodules. Recombination nodules are necessary and sufficient for high levels of sister exchanges, suggesting that SC-CR–mediated coalescence of ZMM proteins promotes all exchanges, whether they occur between homologs or sisters. Finally, we find that sister exchanges occur when the SC-CR assembles both between the sisters and next to them, suggesting that it is proximity to the SC-CR that promotes formation of nearby recombination nodules and, consequently, exchanges.

# Results

## The SC-CR can promote sister exchanges

Previously, we used a novel approach to cytologically score sister exchanges in conditions where the homolog was unavailable, and therefore, the sister was used as the template for homologous recombination. We found that although sister exchanges were rare on the X chromosomes in *him-8* worms—chromosomes that do not pair or associate with SC-CR components—they were common in three scenarios where the SC-CR assembles on unpaired chromosomes: deletion of the cohesin subunit *rec-8*, and mutations in the SC-CR subunits *syp-3(me42)* and *syp-1$^{K42E}$* (Almanzar et al, 2021).

To rule out an X chromosome–specific effect on sister exchanges, we examined *zim-2* animals, where chromosome V does not pair or associate with SC-CR components, in an analogous fashion to the X chromosome in *him-8* mutants (Phillips & Dernburg, 2006). We found sister exchanges on the unpaired chromosomes to be a relatively rare outcome of DNA repair in *zim-2* animals, albeit slightly elevated compared with *him-8* animals: 10.0% versus 4.2% sister exchanges per unpaired chromosome (Fig 1; Pearson's chi-square, *P* = 0.51; see Table S1 for a summary of all sister exchange data). Although not statistically significant, the elevated level of sister exchanges in *zim-2* worms may be caused by a nucleus-wide increase in the number of DSBs (Yu et al, 2016), rather than a chromosome-specific effect on repair. The low rate by which DSBs are resolved as sister exchanges—1–2% (Almanzar et al, 2021)—can account for the mild effect observed in *zim-2* animals. This idea is consistent with elevated sister exchanges on both paired and unpaired chromosomes in *zim-2* animals (see Fig 4D, below), and with the more numerous foci of the DNA repair protein RAD-51—a proxy for the number of DSBs—in *zim-2* versus *him-8* animals (7.87 versus 5.49 foci, *P* < 0.0001, *t* test; Fig S1; [Colaiácovo et al, 2003]).

The conditions we examined previously—*rec-8*, *syp-3(me42)*, and *syp-1$^{K42E}$* animals—mislocalize the synaptonemal complex by eliminating components of the axis (REC-8) or mutating SC-CR components (SYP-1 and SYP-3). To test whether a synaptonemal complex made of unaltered components can also promote sister exchanges, we examined *ieDf2* worms, where all chromosomes fail to pair. In contrast to mutants such as *him-8* or *zim-2*, unpaired chromosomes in *ieDf2* worms undergo widespread "fold-back" that brings together the left and right arms of each chromosome with the SC-CR assembling between them (Harper et al, 2011). We observed 39.0% sister exchanges per chromosome in *ieDf2* worms (Fig 1), significantly higher than in *him-8* animals (*P* = 0.001, Pearson's chi-square).

## Recombination nodules promote sister exchanges

Inter-homolog exchanges, or crossovers, form at recombination nodules, where ZMM proteins accumulate (Pyatnitskaya et al, 2019), leading us to hypothesize that, on unpaired chromosomes, designation of an exchange fate for inter-sister repair events also occurs in recombination nodules.

We first confirmed that the number of recombination nodules, marked by the tagged ZMM protein GFP-COSA-1, corresponds to the number of sister exchanges in the above-mentioned conditions (see Table S1 for a summary of all GFP-COSA-1 counts). In *ieDf2* animals, we observed an average of 4.8 recombination nodules per nucleus, similar to the number of sister exchanges: 4.7 (Fig 2A and B). In *syp-3(me42)* animals, there were 7.4 recombination nodules and 8.0 sister exchanges per nucleus, and in *syp-1$^{K42E}$* animals, there were an average of 16.4 recombination nodules corresponding to 9.6 sister exchanges per nucleus (Fig 2A and B [Almanzar et al, 2021; Gordon et al, 2021]; see the Discussion section for a potential explanation for the latter discrepancy). In *rec-8*

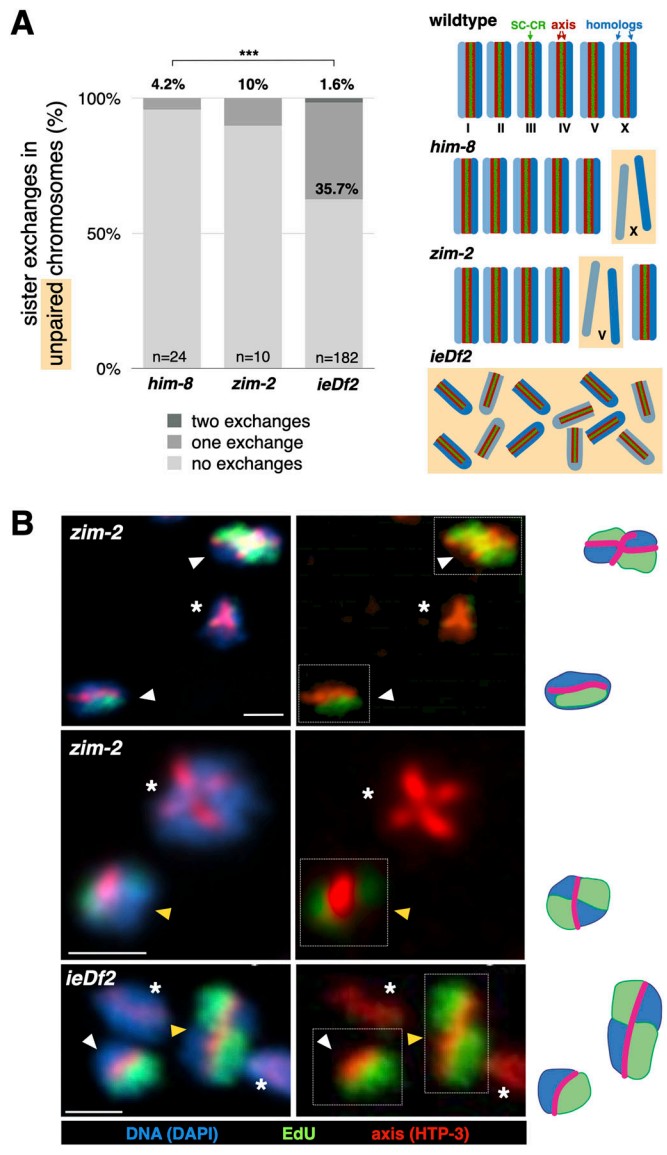

**Figure 1. The SC-CR promotes sister exchanges when it localizes to unpaired chromosomes.**
**(A)** Exchanges are elevated on unpaired chromosomes in *ieDf2* animals, but not in *him-8* and *zim-2* animals. (Data for WT animals are not shown because all chromosomes are paired.) Comparison between *him-8* and *zim-2* was not significant (Pearson's chi-square). Comparison between *him-8* and *ieDf2* animals was significant ($P = 0.001$, Pearson's chi-square). *him-8* data are taken from Almanzar et al (2021). Diagrams of the different genotypes are shown to the right. Chromosomes are shown in blue, the axes in maroon, and the SC-CR in green. Unpaired chromosomes are shown with an orange background. Note that the unpaired chromosomes in *him-8* and *zim-2* animals (the X chromosome and chromosome V, respectively) are not associated with the SC-CR, whereas all chromosomes are unpaired and associated with the SC-CR in *ieDf2* animals. **(B)** Representative images and interpretive diagrams of exchange and non-exchange chromosomes in *ieDf2* and *zim-2* animals. Yellow arrows denote exchange chromatids, white arrows denote non-exchange chromatids, and asterisks denote unlabeled chromatids (not scored). Interpretive diagrams of chromosomes surrounded by dashed white boxes are shown to the right. Red, axis (anti-HTP-3 antibodies); green, EdU; and blue, DNA (DAPI). Note the EdU signal crossing the axis in exchange chromatids. Scale bars = 1 µm.

animals, an average of 10.4 recombination nodules were accompanied by 6.9 sister exchanges per nucleus (Cahoon et al, 2019; Almanzar et al, 2021). In *him-8* animals, consistent with the rare sister exchanges on the unpaired X chromosomes—only 4.2% underwent an exchange—recombination nodules were absent on the X chromosome, and we observed an average of 5.2 recombination nodules per nucleus, one on each of the five autosomes (Fig 2A and B; [Almanzar et al, 2021]).

To test whether recombination nodules are sufficient for promoting sister exchanges, we analyzed worms lacking an SC-CR altogether, a condition where ZMMs are ectopically recruited to sites of DSB repair on unpaired chromosomes (Li et al, 2018; Cahoon et al, 2019). In the absence of the SC-CR (caused by removing the SC-CR components SYP-1 or SYP-2), an average of 3.5 recombination nodules formed on unpaired chromosomes (Fig 2B; see also Li et al, 2018; Cahoon et al, 2019), similar to the number of sister exchanges: an average of 4.6 sister exchanges in *syp-1* and 4.4 in *syp-2* (Fig 2C–E). These numbers are significantly higher than those observed in *him-8*, where recombination nodules do not form on the unpaired chromosomes ($P < 0.01$ in both pairwise comparisons, Pearson's chi-square).

### Most sister exchanges depend on recombination nodules

To test whether sister exchanges require recombination nodules, we used *syp-1*$^{K42E}$ animals grown at 25°C, where sister exchanges are common (Fig 3; [Almanzar et al, 2021]). We conditionally depleted the ZMM factor ZHP-3, which is essential for crossovers, by growing *zhp-3-FLAG-AID* in the presence of auxin (hereafter *zhp-3(–)* [Jantsch et al, 2004; Bhalla et al, 2008; Zhang et al, 2015, 2018]). As expected, *syp-1*$^{K42E}$ *zhp-3(–)* animals completely lacked recombination nodules (Fig 3A) and, compared with *syp-1*$^{K42E}$ animals, underwent significantly fewer sister exchanges: 3.8 versus 9.6 exchanges per nucleus in *syp-1*$^{K42E}$ *zhp-3(–)* and *syp-1*$^{K42E}$, respectively ($P < 0.001$, Pearson's chi-square; Fig 3B and C; the total number of exchanges per nucleus is extrapolated from the rates of single- and double-exchange chromatid, based on 12 chromatids per nucleus). The effect of ZMM depletion suggests that most sister exchanges occur at recombination nodules.

The factors responsible for the remaining sister exchanges in *syp-1*$^{K42E}$ *zhp-3(–)* worms are unknown. ZMM-independent, or "class II," crossovers have been observed in other species (Gray & Cohen, 2016), but their existence and functional relevance in *C. elegans* is a matter of debate (Youds et al, 2010). Based on the lack of chiasmata, interhomolog exchanges are unlikely to form in *zhp-3(–)* or other *zmm* mutants in worms (Jantsch et al, 2004; Yokoo et al, 2012; Zhang et al, 2018). Our finding of sister exchanges forming without recombination nodules in *syp-1*$^{K42E}$ *zhp-3(–)* animals suggests that these exchanges may be generated by the same mechanisms responsible for class II crossovers.

### Sister exchanges are suppressed by crossovers on the same chromosome

One of the most intriguing features of meiotic crossover regulation is the inhibitory effect a crossover exerts on nearby inter-homolog

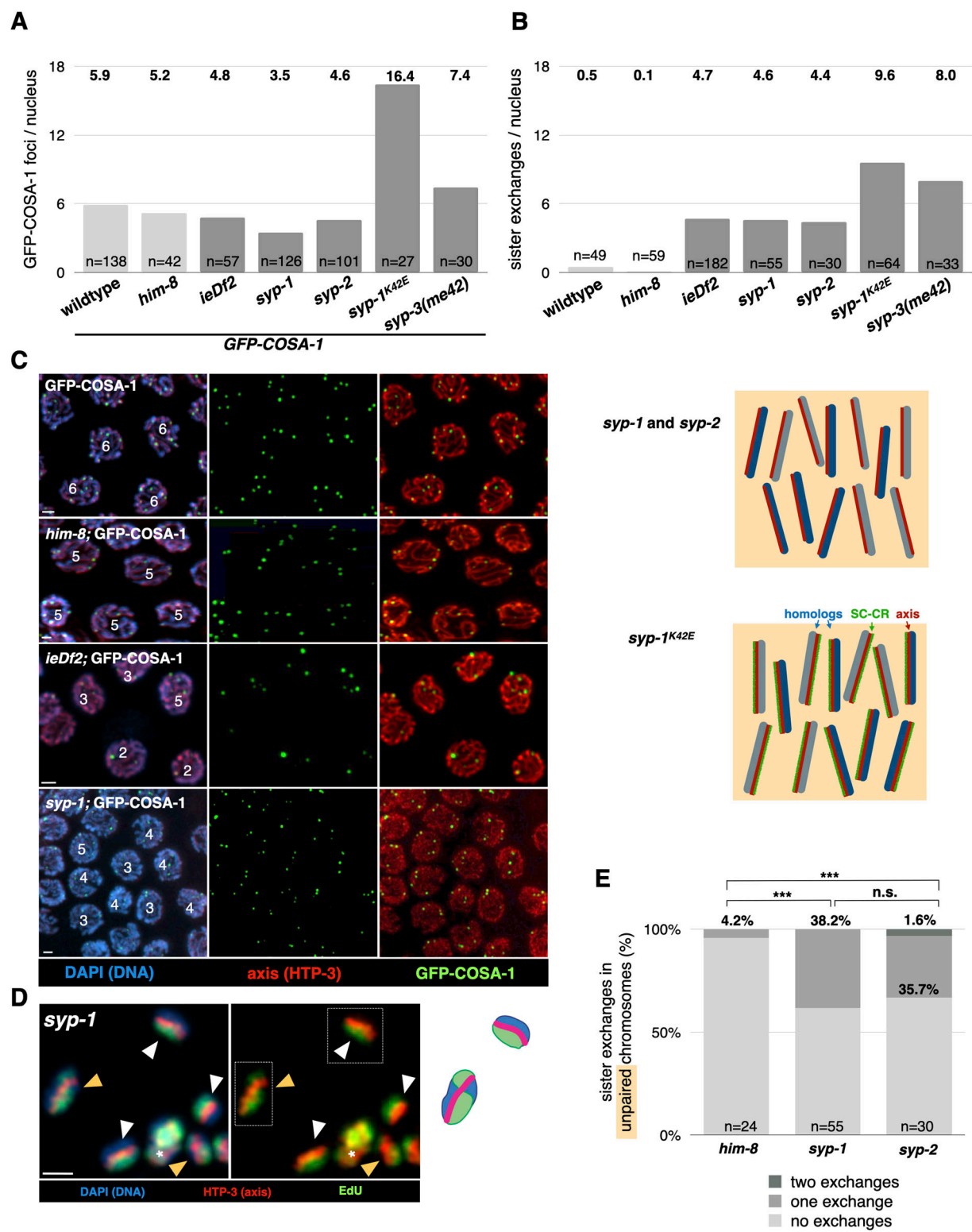

**Figure 2. Sister exchanges correspond to recombination nodules.**
**(A)** Average number of GFP-COSA-1 foci in late-pachytene nuclei of animals of the designated genotype. ("+" indicates animals that have no genetic alterations except for the COSA-1-GFP transgene, which serves as a WT control). Results were significantly different from WT for all genotypes ($P < 0.000001$, $t$ test). n value indicates the number of nuclei counted. Data for *syp-2* are from Cahoon et al (2019); those for *syp-1*[K42E] are from Gordon et al (2021); and those for *syp-3(me42)* are from Almanzar et al (2021). Light gray shading indicates conditions where only zero or two chromosomes are unpaired; dark gray shading indicates all chromosomes are unpaired. **(B)** Total number of sister exchanges extrapolated from the exchange number quantified in Figs 1A and 2D and Almanzar et al (2021). Shading is as in panel (A). Data for *syp-3(me42)*

repair intermediates on the same chromosome (Sturtevant, 1913; Muller, 1916; von Diezmann & Rog, 2021). Termed crossover interference, this regulation is particularly robust in *C. elegans*, where it acts over micron-scale distances to yield exactly one recombination nodule, and therefore crossover, per chromosome (Libuda et al, 2013). Because both crossovers and sister exchanges form in recombination nodules, we wondered whether they interfere with one another.

To test this idea, we analyzed animals that are heterozygous for *nT1*, a large reciprocal translocation between chromosomes IV and V. In *nT1/+* animals, chromosomes IV and V each assemble SC-CR along their entire length but are homologously paired along only 15% of chromosome IV and 39% of chromosome V (MacQueen et al, 2005). It is in this homologously paired region that a ZMM-marked crossover is formed: in *nT1/+* animals, all GFP-COSA-1 foci on chromosome V localized within 1.3 $\mu$m of the 5S locus in the homologously paired region, whereas in animals lacking the translocation, GFP-COSA-1 foci formed all along chromosome V (Fig 4A and B; $P < 0.0001$, $t$ test).

The rest of chromosomes IV and V are juxtaposed with non-homologous sequences, where only sister-directed repair is possible. We found that despite the absence of crossovers in this region, both sister exchanges and the formation of additional recombination nodules remained suppressed: nucleus-wide sister exchanges were not elevated in *nT1/+* animals (Fig 4C and D; compared with WT, *him-8*, or *zim-2*, $P > 0.05$, Pearson's chi-square), and we observed no more than one recombination nodule per chromosome (an average of 6.0 per nucleus; Fig S2). Similar results were observed for worms heterozygous for the large translocation *hT2*, where only 17% of chromosome I and 37% of chromosome III are homologously synapsed. No sister exchanges were observed in 14 chromatids in *hT2/+* animals.

Taken together, these findings suggest that crossovers exert inhibitory effects over sister exchanges *in cis* along the entire length of the chromosome, in a similar manner to the inhibitory effect exerted over additional crossovers on the same chromosome. Importantly, although we did not observe a statistically significant difference in either *nT1/+* or *hT2/+* animals, we cannot rule out minor effects that are below our power of detection.

### The SC-CR promotes sister exchanges in various conformations

During crossover formation, the SC-CR assembles between the two homologs and, consequently, between the two DNA molecules being exchanged (Zickler & Kleckner, 1999). The near-universal conservation of this architecture suggests that the SC-CR may promote exchanges by placing the swapped DNA molecules across

from each other. The cytological evidence for this idea has been mixed, with some data demonstrating a highly stereotypical localization of recombination nodules relative to the SC-CR and the axes (Woglar & Villeneuve, 2018), and others suggesting localization of the recombination nodules near the SC-CR but not exclusively between the axes (Carpenter, 1975; Zickler & Kleckner, 1999). The functional importance of SC-CR and recombination nodule position has also been challenging to test because the SC-CR plays multiple roles in crossover formation, including bringing the homologs together.

Our findings above demonstrate that the ability of the SC-CR to promote recombination nodule formation and inter-homolog exchanges in worms (MacQueen et al, 2002; Jantsch et al, 2004; Zhang et al, 2018) can extend to the promotion of sister exchanges. Because sisters are held together independently of the SC-CR, the ability to cytologically examine sister exchanges allows us to directly test whether the spatial organization of the SC-CR and chromatin plays a part in promoting exchanges.

We used stimulated emission depletion (STED) super-resolution microscopy to probe the conformation of the SC-CR relative to axes, where chromatin is tethered. We first examined WT animals, where the enhanced resolution confirmed that the SC-CR places the axes of the two homologs parallel to one another, separated by 160 nm (Fig 5A; [Goldstein & Slaton, 1982; Köhler et al, 2017]).

We next analyzed *ieDf2* worms, where the SC-CR assembles on unpaired, folded-back chromosomes and promotes sister exchanges. We found a conformation that resembles the WT, with the two axes separated by 150 nm and the space between them occupied by the SC-CR (Fig 5A). In *ieDf2* animals, the SC-CR brings together the left and right arms of the same chromosome (Harper et al, 2011), suggesting that the two sisters being exchanged are situated to one side of the SC-CR. To test this prediction, we localized the 5S rDNA locus relative to the SC-CR using combined immunofluorescence and fluorescence in situ hybridization (FISH; Fig 5B; [Phillips et al, 2009]). We observed two 5S signals per nucleus, each localized to one side of the SYP-2 signal, indicating that in *ieDf2* animals, the sister chromatids are on one side of the SC-CR. In contrast, WT nuclei harbored one 5S FISH signal that extended on both sides of the SC-CR, corresponding to four aligned sisters, two from each homolog (Fig 5C; $P > 0.00001$, Pearson's chi-square).

We examined two other conditions where the SC-CR association with unpaired axes leads to elevated sister exchanges. In *syp-1*$^{K42E}$ animals, we observed the SC-CR and the axis colocalizing to form a single overlapping thread that could not be resolved in STED microscopy (Fig 5A). This suggested that the two exchanged DNA molecules in *syp-1*$^{K42E}$ animals are localized outside the SC-CR. In *rec-8* animals, where each sister forms an axis, the SC-CR assembles

---

are from Almanzar et al (2021). **(A, B)** Note correspondence between the values in panels (A, B) for conditions where all chromosomes are unpaired. **(C)** Representative images of late-pachytene nuclei in WT, *him-8*, *ieDf2*, and *syp-1* animals that also carry the GFP-COSA-1 transgene. Red, axis (anti-HTP-3 antibodies); green, GFP-COSA-1 (anti-GFP antibodies); and blue, DNA (DAPI). Scale bars = 1 $\mu$m. Diagrams of the different genotypes are shown to the right. Chromosomes are shown in blue, the axes in maroon, and the SC-CR in green. Unpaired chromosomes are shown with an orange background. Note that although all chromosomes are unpaired in both scenarios, they are associated with the SC-CR only in *syp-1*$^{K42E}$ animals. **(D)** Representative images of exchange and non-exchange chromatids in *syp-1* animals. Yellow arrows denote exchange chromatids, white arrows denote non-exchange chromatids, and asterisks denote unlabeled chromatids (not scored). Interpretive diagrams of chromosomes surrounded by dashed white boxes are shown to the right. Red, axis (anti-HTP-3 antibodies); green, EdU; and blue, DNA (DAPI). Scale bar = 1 $\mu$m. **(E)** Sister exchanges are elevated upon complete removal of the SC-CR. Pairwise comparisons between *him-8* and *syp-1* or *syp-2* animals were significant ($P = 0.002$ and $P = 0.008$, respectively, Pearson's chi-square test).

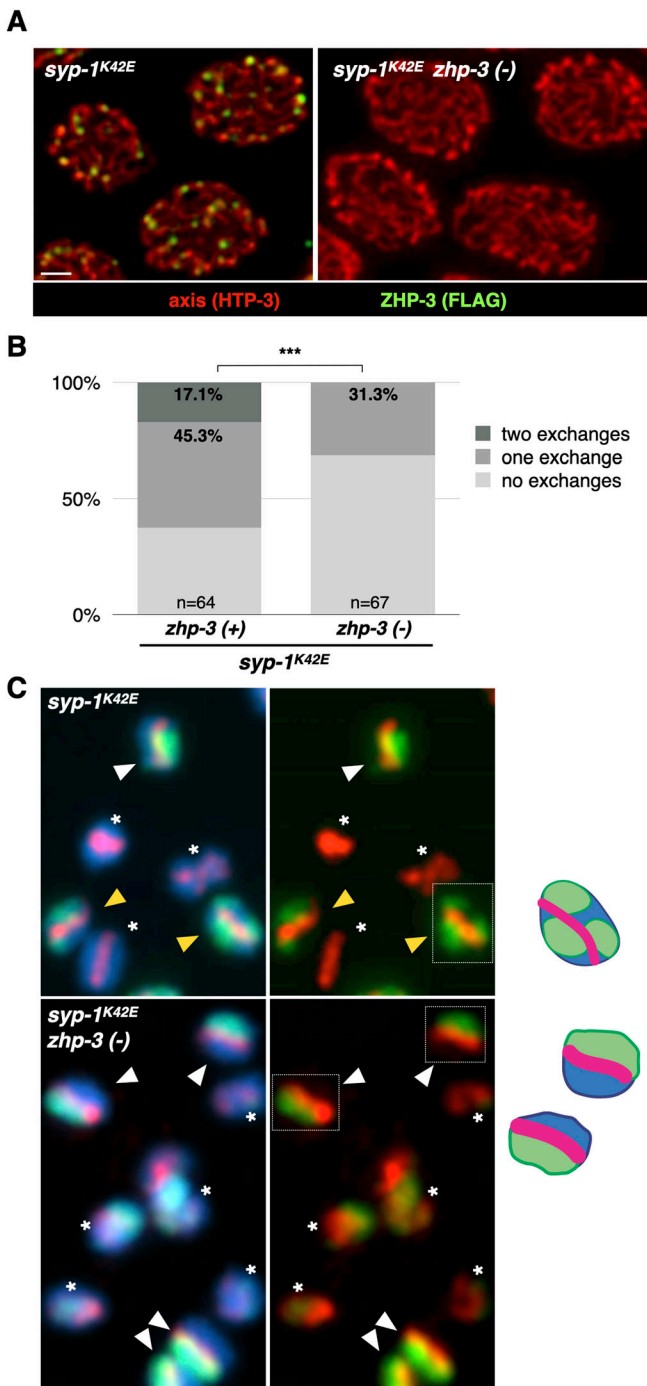

**Figure 3. Most sister exchanges in *syp-1*^K42E animals depend on ZHP-3.**
**(A)** Representative pachytene nuclei in *syp-1*^K42E *zhp-3-FLAG-AID* with and without auxin (right and left, respectively). Note the absence of recombination nodules—foci of ZHP-3—when grown on auxin. Red, axis (anti-HTP-3 antibodies); green, ZHP-3 (anti-FLAG antibodies); and blue, DNA (DAPI). Scale bars = 1 μm. **(B)** Most sister exchanges in the syp-1^K42E background are dependent on ZHP-3. Sister exchanges in *syp-1*^K42E animals grown with or without auxin are significantly different (*P* < 0.001, Pearson's chi-square test). Data for *zhp-3(+)* animals include data from Almanzar et al (2021). **(C)** Representative images of exchange and non-exchange chromosomes in *syp-1*^K42E *zhp-3-FLAG-AID* with and without

between the two sisters, as was previously reported (Fig 5A; [Pasierbek et al, 2001; Cahoon et al, 2019]). The chromosomal architecture in *rec-8* animals is therefore analogous to the WT scenario, with the SC-CR positioned between the DNA molecules undergoing an exchange (a crossover in WT animals or a sister exchange in *rec-8* animals). Notably, and consistent with the myriad roles cohesins play in determining chromosome architecture, the inter-axis distance in *rec-8* was somewhat shorter than in WT or *ieDf2* animals.

The elevated sister exchanges in three conditions with different synaptonemal complex conformations—*syp-1*^K42E, *ieDf2*, and *rec-8*—suggests that the SC-CR does not rely on a specific positioning of the axes or the DNA molecules relative to one another to promote exchanges.

### Recombination nodules form throughout the SC-CR

As an orthogonal way to assess a potential role for the position of the exchanged DNA molecules relative to the SC-CR, we localized recombination nodules relative to the axes. Although previous analysis of hypotonically treated samples suggested that recombination nodules in worms adopt a stereotypical organization in the middle of the SC-CR (Woglar & Villeneuve, 2018), we wanted to localize recombination nodules in three-dimensionally preserved samples.

We visualized GFP-COSA-1 relative to the axes using STED microscopy. In WT animals, where crossovers form at recombination nodules, some foci localized between the axes in the middle of the SC-CR, but many were on one of the axes or off-center (Fig 6). Foci were approximately Gaussian-distributed, with a full width at half maximum (FWHM) of 58% the inter-axis distance (95% CI of 50–68%; N = 94). This variation was ~fourfold greater than the localization precision (Figs 6B and S3A). GFP-MSH-5, part of a meiosis-specific MutSγ complex that binds repair intermediates in vitro (Snowden et al, 2004), exhibited similar distribution (Fig S3B), with FWHM of 58% (95% CI of 52–67%; N = 112; notably, 17% [24/136] of the foci were doublets, which were mostly perpendicular to the axes, as was previously reported [Woglar & Villeneuve, 2018]; these foci were not considered when calculating the distribution).

A similar, though more broadly distributed, localization pattern was observed for *ieDf2* animals, where sister exchanges, rather than inter-homolog crossovers, are generated at recombination nodules. In this case, foci were distributed with a FWHM of 84% of the inter-axis distance (95% CI of 75–97%; N = 112), even though the inter-axis distance was similar to that in WT animals (Fig 5A and see the Materials and Methods section). Notably, 18% of foci localized outside the axes (Fig 6A, bottom, and Fig 6B), significantly more than the 5% observed in WT animals (Fisher's exact test, *P* = 0.025).

auxin. Yellow arrows denote exchange chromatids, white arrows denote non-exchange chromatids, and asterisks denote unlabeled chromatids (not scored). Interpretive diagrams of chromosomes surrounded by dashed white boxes are shown to the right. Red, axis (anti-HTP-3 antibodies); green, EdU; and blue, DNA (DAPI). Scale bars = 1 μm.

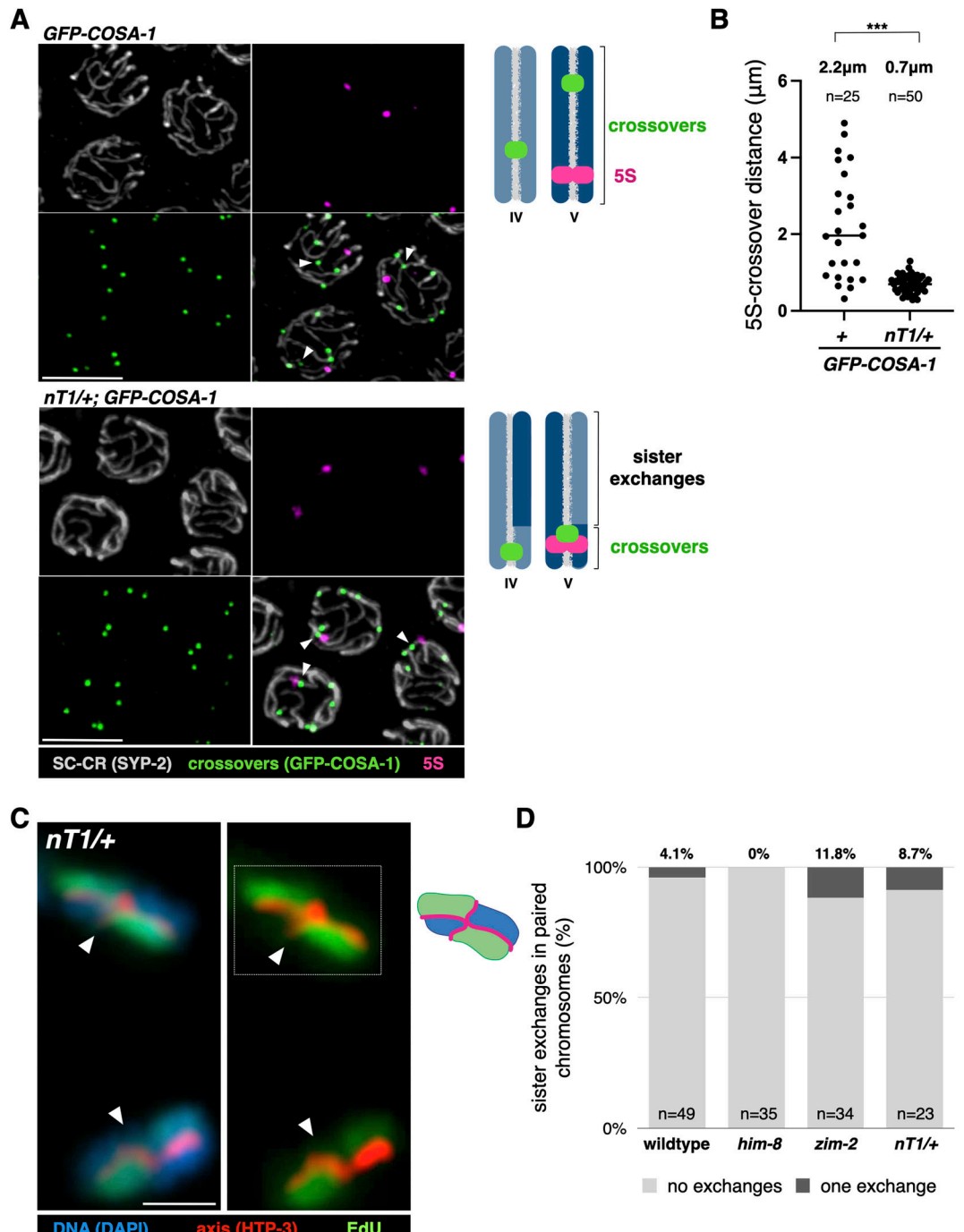

**Figure 4. Sister exchanges and crossovers are regulated together.**
**(A)** Combined immunofluorescence and FISH of pachytene nuclei from *GFP-COSA-1* (control) and *nT1/+; GFP-COSA-1* animals. The 5S locus is located on chromosome V within the homologously-paired portion in *nT1/+*. Gray, SC-CR (SYP-2); green, crossovers (GFP-COSA-1); and magenta, 5S locus. The crossover on chromosome V (judged by tracing the 5S-containing SC-CR signal) is marked with an arrowhead. Chromosomes IV and V are shown in two shades of blue in the diagrams to the right. Scale bars = 5 μm. **(B)** Distance along chromosome V between the 5S locus and the crossover (*P* < 0.0001, *t* test). **(C)** Representative partial projections of non-exchange chromosomes in *nT1/+* animals and an interpretive cartoon. Arrowheads indicate the location of the crossover. Red, axis (anti-HTP-3 antibodies); green, EdU; and blue, DNA (DAPI). Scale bars = 1 μm. **(D)** Sister exchanges remain rare when a majority of chromosomes IV and V can only undergo sister exchanges. Pairwise comparisons of exchanges on paired chromosomes between WT, *him-8*, *zim-2*, and *nT1/+* animals were not significant (*P* > 0.05 for all comparisons, Pearson's chi-square).

Our analysis above suggested that the SC-CR in worms promotes the formation of recombination nodules, and thereby the outcome of repair, independently of the position of the engaged DNA molecules. Consistent with this idea, we found that recombination nodules are not exclusively found in the middle of the SC-CR. Rather, recombination nodule localization is consistent with

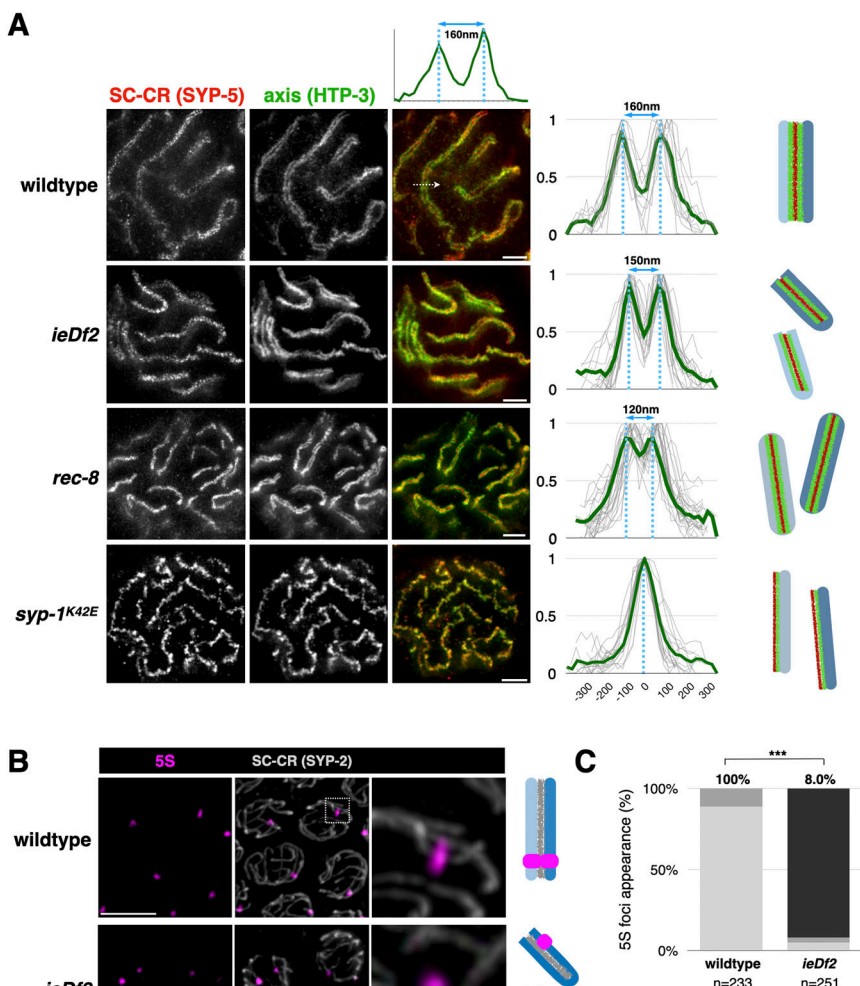

**Figure 5. SC-CR in various conformations can promote sister exchanges.**
**(A)** Representative STED images of pachytene nuclei from WT, *ieDf2*, *rec-8*, and *syp-1*[K42E] worms. Red, SYP-5; and green, HTP-3. Scale bars = 1 μm. HTP-3 fluorescence is normalized such that maximum fluorescence is 1. Dashed white arrow denotes line scan in the merged WT image, with the normalized fluorescence plot shown above. The SC-CR localizes between the axes in WT, *ieDf2*, and *rec-8* worms, but the SC-CR and axis signals colocalize in *syp-1*[K42E] worms. Overlaid line scans show similar distribution in WT (n = 12) and *ieDf2* (n = 27) worms—two axis peaks separated by 150–160 nm. Rec-8 animals (n = 27) exhibit a somewhat smaller inter-axis distance—120 nm—whereas *syp-1*[K42E] worms (n = 18) show a single peak. Averages are shown in green. Diagrams of chromosome architecture are shown to the right, with the two homologs shown in two shades of blue.
**(B)** Representative confocal images of pachytene nuclei from WT and *ieDf2* animals. Gray, SC-CR (SYP-2); and magenta, 5S locus. Diagrams of chromosome architecture are shown to the right, with the homologous chromosome V shown in two shades of blue. Because the homologs are not paired in *ieDf2*, each nucleus harbors two 5S foci.
**(C)** In WT animals, foci span the SC-CR, indicating the four sisters (two from each homolog) are aligned. In contrast, most foci in *ieDf2* animals are to one side of the SC-CR (*P* > 0.00001, Pearson's chi-square).

their formation throughout the volume of the SC-CR or at its top or bottom surfaces.

## Discussion

Our work demonstrates that recombination nodules can promote sister exchanges. We document correspondence between the number of recombination nodules and sister exchanges in multiple independent scenarios (Figs 1 and 2), dependency of most sister exchanges on recombination nodules (Fig 3), and regulatory interplay between crossovers and sister exchanges (Fig 4). These data indicate that although only crossovers were thus far known to form at recombination nodules, the same molecular machinery can also regulate sister exchanges. Although several repair pathways and chromosomal structures have been suggested to regulate meiotic sister exchanges (e.g., Adamo et al, 2008), only a few have direct experimental evidence. In addition to regulation by recombination nodules reported here, these include the BLM helicase, which

exhibits an anti-recombination activity (Jessop & Lichten, 2008; Oh et al, 2008; Almanzar et al, 2021); and factors that bias repair toward the homolog rather than the sister (Humphryes & Hochwagen, 2014): the cohesin complex, which limits sister exchanges in flies, yeast, and worms (Webber et al, 2004; Kim et al, 2010; Almanzar et al, 2021), and the budding yeast meiotic kinase Mek1 (Hollingsworth & Gaglione, 2019).

An important implication of our finding is that the execution of the exchange/non-exchange decision in worms likely relies on common features of inter-homolog and inter-sister events—for example, repair intermediates like double Holliday junctions (Schwacha & Kleckner, 1995)—rather than on their differential organization within meiotic chromosomes. Although inter-homolog–specific processing would suggest that recombination nodules would be positioned strictly between the two homologs (Woglar & Villeneuve, 2018, but see Carpenter, 1975; Zickler & Kleckner, 1999), we find that recombination nodules do not localize at a strict stereotypical location relative to the synaptonemal complex or to the two DNA molecules undergoing an exchange

⬥⬥⬥ **Life Science Alliance**

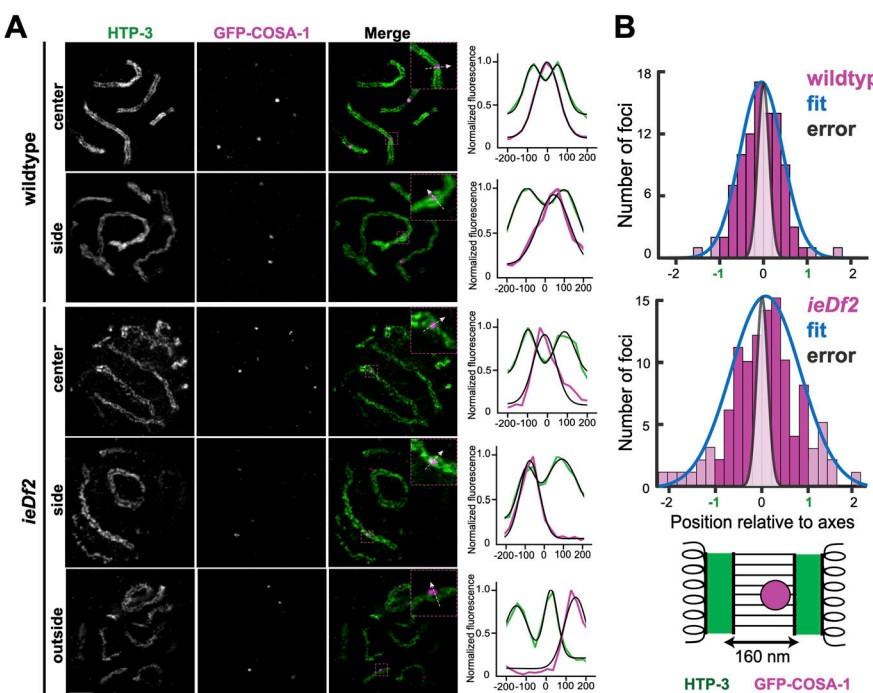

**Figure 6. Recombination nodules do not form exclusively in the middle of the SC-CR.**
**(A)** STED images of nuclei from the WT and *ieDf2* worms immunolabeled for GFP-COSA-1 (magenta) and HTP-3 (green). Right, intensity distribution extracted from line profiles of merged images (insets). Black lines denote fits to either one (GFP-COSA-1) or a sum of two (HTP-3) Gaussian distribution plus constant background. Scale bars = 1 μm. **(B)** Histograms of GFP-COSA-1 focus center positions for WT (n = 94) and *ieDf2* (n = 112). Values of 1 or −1 indicate localization on the meiotic axes, whereas a value of 0 indicates localization at the center of the SC-CR. Magnitudes >1 or <−1 (lighter bars) indicate GFP-COSA-1 center position outside the axes. Blue curves represent a Gaussian fit to the data, whereas the gray distribution represents estimated localization error (Fig S3); for clarity, curves are displayed with the same maximum value as each histogram. Bottom, schematic diagram of synaptonemal complex geometry, with axes marked in green and the GFP-COSA-1 focus in magenta.

(Figs 5 and 6). Although the multiple, inter-dependent meiotic roles of the synaptonemal complex have complicated mechanistic probing, our findings suggest that it regulates recombination nodule formation by a proximity-based mechanism.

## Regulation of crossovers and sister exchanges

Our work supports a two-tiered mechanism regulating homologous recombination during *C. elegans* meiosis. First, homolog bias directs most repair events to engage the homolog as a template. It is only in the absence of the homolog that the repair machinery resorts to using the sister chromatid as a template (MacQueen et al, 2005). A second independent layer carries out the exchange/non-exchange decision: although most of the events form non-exchanges, those repair events designated by recombination nodules avoid this default fate and form exchanges. Together, these two tiers tightly regulate crossovers to ensure accurate chromosome segregation while channeling all other sister- and homolog-directed events into a non-exchange fate.

An open question is whether limiting the number of meiotic sister exchanges has functional importance for successful meiosis. Non-exchanges between sisters may be preferred as a consequence of the structural similarity between inter-sister and inter-homolog repair intermediates (Sanchez et al, 2021). The tight regulation on the number of crossovers may, as a by-product, channel inter-sister events into the default, non-exchange fate. A non-mutually exclusive idea is that sister exchanges are deleterious, especially when occurring in regions rich in sequence repeats (Vader et al, 2011), and actively avoiding them helps protect genome integrity in the germline.

Our data indicate many commonalities in the SC-CR–mediated regulation of crossovers and sister exchanges—among them, correspondence with the number of recombination nodules, localization relative to the axes, and co-regulation *in cis* on the same chromosome. However, it also points to potential differences. These include the localization of a minority of inter-sister recombination nodules outside the SC-CR (Fig 6), and the seemingly lower efficiency of forming recombination nodules in *ieDf2* animals, where only four of the 12 SC-CR stretches in the nucleus harbor a recombination nodule (Figs 1 and 2). In addition, minor differences in exchange efficiency between crossovers and sister exchanges may be revealed by future work to measure the number of DSBs in worms—currently a matter of debate because the available proxies, such as RAD-51 foci, are indirect and conflate DSB number with repair dynamics (Yu et al, 2016). Although we have mitigated this limitation in our experiments by comparing scenarios where meiotic checkpoints are already activated (rather than comparing perturbed and WT meioses), this knowledge gap prevented us from defining an exchange frequency per sister-directed repair event in the different scenarios we have analyzed.

## Regulation of recombination nodules by the SC-CR

Our data show that the SC-CR plays a crucial role in recruiting ZMM proteins and promoting the formation of recombination nodules. Some ZMM factors have an intrinsic affinity for the SC-CR: the ZMM proteins ZHP-1-4 and Vilya in worms and flies, respectively, localize along the SC-CR before coalescing into recombination nodules. ZMM factors also localize to chromatin-free assemblies of SC-CR

material (called polycomplexes) in worms, flies, and yeast (Tsubouchi et al, 2006; Lake et al, 2015; Shinohara et al, 2015; Rog et al, 2017; Zhang et al, 2018; Voelkel-Meiman et al, 2019). Our work suggests that in worms, the recruitment of ZMM proteins to the SC-CR and their coalescence into recombination nodules promote the formation of exchanges independently of the role of the SC-CR in bringing homologs together. We propose that SC-CR–mediated recruitment acts in concert with the affinity of ZMM members, such as the MutSɣ complex Msh4-5 or the Zip2-Zip4-Spo16 complex, for specific repair intermediates (Snowden et al, 2004; De Muyt et al, 2018). Although the molecular details of SC-CR–ZMM interactions in worms are unknown, recent work in budding yeast identified an interaction interface between the ZMM protein Zip4 and the SC-CR component Ecm11 (Pyatnitskaya et al, 2022).

Surprisingly, we find that the conserved conformation of the synaptonemal complex relative to the chromosomes is not necessary for its ability to promote the formation of recombination nodules (Fig 5). In addition, rather than being located in the middle of the SC-CR, recombination nodules were distributed throughout its width (Fig 6). This suggests that the SC-CR may act to initially concentrate ZMM proteins, and later to create a local environment in its vicinity that is conducive to their coalescence into recombination nodules. Consistent with this idea is the location of dense recombination nodules outside the SC-CR in electron micrographs of meiotic chromosomes in many species (Carpenter, 1975; Zickler & Kleckner, 1999), and foci resembling recombination nodules that form by ZMM proteins at the periphery of polycomplexes in yeast and worms (Tsubouchi et al, 2006; Shinohara et al, 2015; Rog et al, 2017; Zhang et al, 2018; Voelkel-Meiman et al, 2019).

Notably, recombination nodules in some organisms, including mammals and budding yeast, nucleate SC-CR assembly (Pyatnitskaya et al, 2019). This function of recombination nodules—promoting SC-CR assembly—likely involves stereotypic placement of the axes relative to one another. However, it is unknown whether in these organisms the role we study in this work—regulating exchanges—also relies on specific conformation of recombination nodules vis-à-vis the axes and the SC-CR. Because the exchange-promoting role is highly conserved, it is possible that recombination nodules in mammals and budding yeast, like in worms, do not rely on precise orientation relative to the axes and the SC-CR to regulate exchanges.

In addition to proximity-based positioning of recombination nodules, the SC-CR may play additional roles in regulating meiotic exchanges. In *ieDf2* worms, where the DNA molecules undergoing exchange were to one side of the SC-CR (Fig 5), a minority of recombination nodules localized outside the SC-CR (Fig 6). This points to a potential role for the conformation of the SC-CR relative to chromatin (or for factors that depend on this organization) in the tight regulation of recombination nodule position. Similarly, the presence of more recombination nodules than sister exchanges in *syp-1$^{K42E}$* animals (Figs 1 and 2 and Almanzar et al, 2021) hints at a potential role of SC-CR conformation relative to the axes in ensuring the fidelity of exchange designation by recombination nodules.

What kind of material properties of the SC-CR and the ZMM proteins could support these complex dynamics observed in worms—initial colocalization followed by the formation of juxtaposed yet distinct entities? The discovery that the SC-CR is a liquid-like compartment provides a potential mechanism (Rog et al, 2017). ZMM proteins initially interact more loosely with the SC-CR, either diffusing within it or forming a film around it. Coalescence of dynamic ZMM proteins into a recombination nodule reflects the formation of a compartment with biophysical properties distinct from the SC-CR (Jantsch et al, 2004; Bhalla et al, 2008; Rog et al, 2017; Zhang et al, 2018). The coalescence and location of the recombination nodule compartment may be regulated by wetting or differential miscibility with the SC-CR (Feric et al, 2016; Wan et al, 2018). Repair intermediates, for which certain ZMM proteins have an affinity (Snowden et al, 2004; De Muyt et al, 2018), may direct the position of these coalescing events; in turn, these repair intermediates may be repositioned by forces exerted by interactions between the two compartments (Shin et al, 2018). Future work probing the three-dimensional organization and dynamics of recombination nodules, and the potential conservation of these properties beyond worms, will test these mechanisms.

# Materials and Methods

### Worm strains and growing conditions

*C. elegans* worms were generated and cultured using standard conditions and protocols (Brenner, 1974). Worms were grown at 20°C, except worms carrying the *syp-1$^{K42E}$* allele, which were maintained at 15°C, but were grown at 25°C from hatching before experimentation. Auxin treatment was conducted as in Almanzar et al (2021) and Zhang et al (2015). *ieDf2* is a deletion of *him-8*, *zim-1*, *zim-2*, and *zim-3* (Harper et al, 2011). The *ieDf2*, *syp-1*, and *syp-2* alleles were maintained as balanced strains, with homozygous progeny picked before experimentation. *syp-1(me17)* balanced by *nT1* was used for the *nT1/+* experiments. *htp-3(tm3655)* balanced by *hT2* was used for the *hT2/+* experiments.

### Sister exchange analysis

EdU treatment, click chemistry, and sister exchange quantification were conducted as in Almanzar et al (2021, 2022). The total number of sister exchanges (Fig 2B and Table S1) was extrapolated from the exchange rates on paired and unpaired chromosomes based on six paired chromosomes in WT, *nT1/+*, and *hT2/+* animals, five paired and two unpaired chromosomes in *him-8* and *zim-2* animals, and 12 unpaired chromosomes in *ieDf2*, *rec-8*, *syp-1*, *syp-2*, *syp-3(me42)*, and *syp-1$^{K42E}$* animals.

### Confocal microscopy

Immunofluorescence was performed as previously described (Phillips et al, 2009; Gordon et al, 2021), using ProLong Glass Antifade Mountant. Combined FISH and immunofluorescence experiments were performed essentially as in Phillips et al (2009). 5S

probe was amplified using primers 5′-TACTTGGATCGGAGACGGCC-3′ and 5′-CTAACTGGACTCAACGTTGC-3′. The resulting ~1-kb PCR product was digested with MluCI (isoschizomer of Tsp509I; New England Biolabs) and fluorescently labeled using the Ulysis Alexa Fluor 647 Nuclei Acid Labeling Kit (Thermo Fisher Scientific). All confocal microscopy images (Figs 1–4, 5B, S1, and S2) were taken using a Zeiss LSM 880 confocal microscope equipped with an AiryScan and using a 63× 1.4 NA oil immersion objective. Images were processed using Zen Blue 3.0. Partial maximum-intensity projections are shown throughout.

## STED microscopy

Gonads were stained as previously reported (Phillips et al, 2009; Gordon et al, 2021) with the following modifications: for fixative, we used a final concentration of 2% formaldehyde diluted from 16% ampules opened immediately before dissection; in addition, DAPI staining was omitted and MOUNT LIQUID (Abberior) was used as mounting media. STAR RED and Alexa Fluor 594 were used for the two STED channels (see the Antibodies section, below). Images were acquired with an Abberior STEDYCON mounted on a Nikon Eclipse Ti microscope with a 100× 1.45 NA oil immersion objective. Images shown are a single z-section.

## Antibodies

The following antibodies were used: guinea-pig anti-HTP-3 (1:500; [Hurlock et al, 2020]), rabbit anti-SYP-5 (1:500; [Hurlock et al, 2020]), rabbit anti-SYP-2 (1:500; [Hurlock et al, 2020]), rabbit anti-RAD-51 (1:10,000; [Harper et al, 2011]), mouse anti-GFP (1:2,000; Roche), mouse anti-FLAG (1:500; Sigma-Aldrich), Cy3 AffiniPure donkey anti-guinea-pig (1:500; Jackson Immuno-Research), 488 AffiniPure donkey anti-mouse (1:500; Jackson ImmunoResearch), 488 AffiniPure donkey anti-rabbit (1:500; Jackson ImmunoResearch), 594 AffiniPure donkey anti-guinea-pig (1:200; Jackson ImmunoResearch), 594 AffiniPure donkey anti-mouse (1:200; Jackson ImmunoResearch), goat anti-rabbit STAR RED (1:200; Abberior), goat anti-mouse STAR RED (1:200; Abberior), and donkey anti-guinea-pig STAR RED (1:200; Abberior).

## Quantification of RAD-51 and GFP-COSA-1 foci

Gonads stained with anti-RAD-51 antibodies was imaged. Maximum-intensity projections spanning whole nuclei were set to the same threshold values across all genotypes to remove background staining. Foci from nuclei in mid-pachytene were counted. At least three gonads in each genotype were counted. GFP-COSA-1 was quantified by staining with an anti-GFP antibody, and scoring late-pachytene foci, as previously reported (Yokoo et al, 2012; Almanzar et al, 2021).

## Quantification of inter-axis width

Line scans across chromosomes in frontal view, where the axes and the SC-CR were in the xy-plane, were performed using ImageJ. After background subtraction (using the pixel with the lowest signal along the line scan), fluorescence measurements were normalized so that the pixel with the highest fluorescence is set to 1. Fluorescence plots were manually aligned so that the inter-axis minimum (for WT, *ieDf2 and rec-8*) or maximum (for *syp-1*$^{K42E}$) was at the 0 $\mu m$ point.

## Quantification of FISH data

Late-pachytene nuclei were analyzed. At least three gonads were analyzed for each condition. Localization of the 5S relative to the SC-CR was done by analyzing foci throughout multiple z-slices. Distance between 5S and GFP-COSA-1 foci was analyzed in Imaris (Bitplane). The distance along the SC-CR was calculated using the Measurement Point tool.

## Quantification of GFP-COSA-1 foci position relative to the SC-CR

To estimate the position of recombination nodules, we drew line profiles of the fluorescence intensity of GFP-COSA-1 and HTP-3, using the position of SYP-2, which localizes in the center of the SC-CR (Fig S3; [Köhler et al, 2020, *Preprint*]), as an upper bound of our localization precision. Fitting of GFP-COSA-1 and SYP-2 position relative to HTP-3 axis locations was performed using custom scripts written in MATLAB (scripts available upon request; The MathWorks). Briefly, line profiles were generated from STED data using the line profile function of Fiji (Schindelin et al, 2012) with a three-pixel wide (~75 nm) line average. Fits to each line profile were performed using the lsqnonlin function of MATLAB, minimizing the residuals between the line profile data $y(x)$ and a

$$y_{model}(x) = A + B \exp\left(-\frac{(x-C)^2}{2D}\right) + E \exp\left(-\frac{(x-F)^2}{2G^2}\right)$$

for free parameters A–G (or A–D for a single Gaussian). Initialization of the fit was performed by finding local maxima of $y(x)$. To avoid overfitting of background for the double Gaussian, only data within 160 nm of either local maximum were fit. GFP-COSA-1 (or SYP-2) positions were normalized by subtracting the midpoint position of the two fit axis positions and dividing by the inter-axis distance of that profile; the sign of the distance was selected randomly for each measurement. Measured inter-axis distances were similar in GFP-COSA-1 experiments in WT and *ieDf2* worms, and in SYP-2 experiments, with mean values of 161 ± 25 nm (N = 94), 156 ± 27 nm (N = 112), and 162 ± 17 nm (N = 10), respectively (errors: SD). Distribution of GFP-COSA-1 and SYP-2 positions within the inter-axis coordinate system was fit using maximum likelihood estimation to the Gaussian distribution, with FWHM of 0.58 (0.50, 0.68), 0.85 (0.73, 1.02), and 0.15 (0.11, 0.29) relative to the distance between axes (ranges: 95% confidence interval).

## Statistical analysis

Chi-square and *t* tests were conducted using a combination of the RStudio software package and GraphPad Prism. For tests involving sister exchanges, single- and double-exchange chromatids were pooled together.

**Summary of resources and reagents.**

| Reagent or Resource | Source | Identifier |
|---|---|---|
| Antibodies | | |
| Guinea-pig anti-HTP-3 | Yumi Kim Lab | n/a |
| Rabbit anti-SYP-2 | Yumi Kim Lab | n/a |
| Rabbit anti-SYP-5 | Yumi Kim Lab | n/a |
| Rabbit anti-RAD-51 | Abby Dernburg lab | n/a |
| Mouse anti-GFP | Roche | #11814460001 |
| Mouse anti-FLAG | Sigma-Aldrich | #F1804 |
| 488 AffiniPure donkey anti-mouse | Jackson ImmunoResearch | Cat#715-545-150; RRID: AB_2340846 |
| 488 AffiniPure donkey anti-rabbit | Jackson ImmunoResearch | Cat#711-545-152; RRID: AB_2313584 |
| Cy3 AffiniPure donkey anti-guinea-pig | Jackson ImmunoResearch | Cat#706-165-148; RRID: AB_2340460 |
| 594 AffiniPure donkey anti-guinea-pig | Jackson ImmunoResearch | Cat#706-585-148; RRID: AB_2340474 |
| 594 AffiniPure donkey anti-mouse | Jackson ImmunoResearch | Cat#715-585-150; RRID: AB_2340854 |
| Goat anti-mouse STAR RED | Abberior | STRED-1001 |
| Goat anti-rabbit STAR RED | Abberior | STRED-1002 |
| Goat anti-guinea-pig STAR RED | Abberior | STRED-1006 |
| Chemicals, peptides, and recombinant proteins | | |
| EdU | Thermo Fisher Scientific | #A10044 |
| ProLong Glass Antifade Mountant | Thermo Fisher Scientific | #P36980 |
| MOUNT LIQUID | Abberior | #MM-2007 |
| 37% Formaldehyde | Alfa Aesar | #14835 |
| 16% Formaldehyde (methanol-free) | Thermo Fisher Scientific | #28906 |
| Tetramisole | Sigma-Aldrich | #5086-74-8 |
| Roche blocking powder | Roche | #11096176001 |
| MluCI | New England Biolabs | #R0538 |
| Critical commercial assays | | |
| Click-it EdU Cell Proliferation Kit | Thermo Fisher Scientific | #C10337 |
| Ulysis Alexa Fluor 647 Nuclei Acid Labeling Kit | Thermo Fisher Scientific | #U21660 |
| Experimental models: organisms/strains | | |
| *C. elegans*: N2 | CGC | N2 |
| *him-8(tm611) IV* | CGC | CA257 |
| *meIs8 [pie-1p::GFP::cosa-1 + unc-119(+)] II* | CGC | AV620 |
| *ieDf2 [unc-119+]/mIs11 [myo-2p::GFP + pes-10p::GFP + F22B7.9::GFP] IV* | CGC | CA998 |
| *meIs8 [pie-1p::GFP::cosa-1 + unc-119(+)] II; ieDf2 [unc-119+]/mIs11 [myo-2p::GFP + pes-10p::GFP + F22B7.9::GFP] IV* | This study | ROG376 |
| *zim-2(tm574) IV* | CGC | CA258 |
| *meIs8 [pie-1p::GFP::cosa-1 + unc-119(+)] II; syp-1(me17) V/ nT1[unc-?(n754) let-? qIs50] IV;V* | This study | ROG377 |
| *meIs8 [pie-1p::GFP::cosa-1 + unc-119(+)] II; pSUN-1::TIR-1:: mRuby IV; him-8(tm611) spo-11::AID IV* | Gordon et al (2021) | ROG130 |
| *zhp-3(ie76[zhp-3::AID::3xFLAG]) I; unc-119(ed3) III; ieSi38 [sun-1p::TIR1::mRuby::sun-1 3'UTR, Cbr-unc-119(+)] IV; syp-1(slc11) [K42E] V* | This study | ROG385 |
| *syp-1(me17) V/nT1[unc-?(n754) let-? qIs50] IV;V* | CGC | AV307 |
| *syp-2(ok307) V/nT1[unc-?(n754) let-? qIs50] IV;V* | CGC | AV276 |

**Continued**

| Reagent or Resource | Source | Identifier |
|---|---|---|
| syp-1(slc11) [K42E] V | Gordon et al (2021) | ROG198 |
| meIs8 [pie-1p::GFP::cosa-1 + unc-119(+)] II; syp-1(slc11) [K42E] V | Gordon et al (2021) | ROG202 |
| msh-5[ddr22(GFP::msh-5)] IV | Janisiw et al (2018) | NSV129 |
| htp-3(tm3655) I/hT2 [bli-4(e937) let-?(q782) qIs48] I;III | CGC | TY5038 |
| Oligonucleotides | | |
| 5S for: 5′-TACTTGGATCGGAGACGGCC-3′ | IDT | |
| 5S rev: 5′-CTAACTGGACTCAACGTTGC-3′ | IDT | |
| Software and algorithms | | |
| Zen Blue 3.0 | Zeiss | |
| Zen Black 2.3 | Zeiss | |
| Fiji | https://imagej.nih.gov/ij/ | |
| RStudio version 4.2.0 | https://www.rstudio.com/ | |
| GraphPad Prism version 9.4.0 | https://www.graphpad.com | |
| MATLAB | The MathWorks | |
| Imaris 9.9 | Bitplane | |

# Supplementary Information

# Acknowledgements

We would like to thank members of the Rog Lab and Kent Golic for critical reading of this article; Sara Nakielny for comments on the article and editorial work; the University of Utah Cell Imaging Facility for STED microscopy resources; and Yumi Kim and Abby Dernburg for antibodies. Worm strains were provided by Nichola Silva and the CGC, which is funded by the NIH Office of Research Infrastructure Programs (P40 OD010440). L von Diezmann is The Mark Foundation for Cancer Research Fellow of the Damon Runyon Cancer Research Foundation (DRG-2372-19). This work was supported by a Genetics Training Grant T32GM007464 to DE Almanzar, and by a Pilot Project Award from the American Cancer Society, R35GM128804 grant from NIGMS, and start-up funds from the University of Utah to O Rog.

## Author Contributions

DE Almanzar: conceptualization, resources, formal analysis, validation, investigation, visualization, methodology, and writing—original draft.
SG Gordon: resources, validation, investigation, visualization, and methodology.
C Bristow: resources, validation, investigation, visualization, and methodology.
A Hamrick: resources, validation, investigation, visualization, and methodology.
L von Diezmann: resources, software, and supervision.
H Liu: resources.
O Rog: conceptualization, formal analysis, supervision, visualization, project administration, and writing—original draft, review, and editing.

## Conflict of Interest Statement

The authors declare that they have no conflict of interest.

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
