## [Reviewer comments · Life Science Alliance]

Life Science Alliance

Meiotic DNA exchanges in *C. elegans* are promoted by proximity to the synaptonemal complex

David Almanzar, Spencer Gordon, Chloe Bristow, Antonia Hamrick, Lexy von Diezmann, Hanwenheng Liu, and Ofer Rog
DOI: <https://doi.org/10.26508/lsa.202301906>

Corresponding author(s): Ofer Rog, University of Utah

Review Timeline:

Submission Date:	2023-01-06
Editorial Decision:	2023-01-09
Revision Received:	2023-01-09
Editorial Decision:	2023-01-10
Revision Received:	2023-01-10
Accepted:	2023-01-11

Transaction Report:

Please note that the manuscript was previously reviewed at another journal and the reports were taken into account in the decision-making process at Life Science Alliance.

Referee #1 Review

Report for Author:

Dr. Rog,

"Meiotic DNA exchanges in *C. elegans* are promoted by proximity to the synaptonemal complex".

This is a re-review of the revised manuscript for a previous submission by the authors.

This paper by Almanzar shows a connection of chromosome structure compartment with sister chromatid (SCE) in *C. elegans* meiosis. The authors analyzed SCE formation in various mutants and found that SCE formation relies on the assembly of components of the central region of the synaptonemal complex (SC), a meiosis-specific ladder-like chromosome structure that juxtaposes two homologous axes, which would provide a nuclear compartment (domain) for the assembly of recombination nodules (RNs) containing pro-crossover proteins. The experiments were carried out with good quality and most of the results described are solid.

However, in terms of our understanding of the molecular mechanism of meiotic recombination including SCE, the results (and the proposal) are almost experimental confirmation of the previous ideas. Among them, the following are new. First, the authors found the RN in the region where SCE occurs (1-to-1 correspondence). Second, inter-homolog recombination would communicate with inter-sister recombination, manifested by interference. The latter idea could be extended by studying the super-resolution structure of synapsed translocated chromosomes in nT1/+ strain (Figure 4B)-how SC central regions are assembled on the chromosomes such as homologous synapsed region and non-homologous synapsed regions. Alternatively, as shown in *Arabidopsis thaliana* (Durand et al. *Nature Commun*, 2022), in respect of coarsening in RNs in the SC compartment, it would be interesting to see the gene dosage effect of the pro-crossover factor such as the overexpression on the distribution/number of RNs and/or SCEs along chromosomes, although this is a bit out of the topic in the paper.

In conclusion, the revision is not enough to change the previous conclusion by this reviewer that, without new data sets, this paper shows incremental progress in our understanding of the molecular link of meiotic recombination to the chromosome structure/compartment.

Referee #2 Review

Report for Author:

Revision assessment: The authors were able to address most of the concerns from the first set of reviews, making this revision now suitable for publication. In addition to clarifying some of the figures, the authors have provided additional experiments to address concerns by the first set of reviews (addressing interference, conformation of the sisters in ieDf2, and repair intermediate localization in regard to the central region). Furthermore, they have now included appropriate citations and work from other model systems and discuss similarities and differences between model organisms. The only concern to be raised is perhaps in the wording of the conclusion that the SC regulates SCEs. While the SC can promote SCEs when built between sisters, the SC would normally not regulate their occurrence but block those exchanges from ever occurring since it forms between homologs. In addition, these studies could be addressing the processes that prevent SC from forming between sisters and not the regulation per se of SCEs.

Referee #3 Review

Report for Author:

In their revision, Almanzar et al. have addressed many of the concerns expressed in previous reviews, and the manuscript has been improved accordingly. The manuscript remains somewhat bifurcated, with the first part containing an exploration of factors that contribute to sister chromatid exchange (SCE) and formation of recombination nodules (RNs) involving sister chromatids. The second part examines the location of RNs relative to axes and synaptonemal complex central elements in either homologously or non-homologously synapsed chromatids. While both parts of the manuscript contain worthwhile contributions to the literature, neither alone or in combination provide conclusive support to the conclusion regarding the relationship between crossovers and the synaptonemal complex stated in the title or abstract.

Comments (referral is to line numbers):

61-63. This sentence is incomplete.

122 and ff. Why so much discussion of zim-2 and him-8 when SCE levels in the two are not statistically different?

174 and ff. ZHP-3 depletion in syp-1K42E animals results in a 2.5-fold reduction in SCE, but a complete loss of RNs. However, unlike in other SCE-promoting mutants/variants, where SCEs and RNs are in roughly equal numbers, syp-1K42E mutants display a 2-fold excess of RNs over SCEs, suggesting that the two are already uncoupled in this mutant. It would also be useful to compare the 2.5-fold reduction in SCE in auxin-depleted ZHP-3 syp-1K42E mutants with the fraction of interhomolog crossovers (i.e. bivalents at diplotene/diakinesis) remaining upon auxin-depletion of otherwise wild-type cells either 212 and ff. SCEs in nT1/+ animals are not significantly different from those in WT, him-8 or zim-2, but the statistical power to detect differences is not stated. This is important for the assessment of to what extent an interhomolog crossover could possibly affect SCE.

334 and ff. While Woglar & Villeneuve may have suggested a stereotypical RN positioning relative to the SC-CR, previous cytological data have suggested a broader distribution.

415 and ff. An alternative explanation for the broader distribution of RNs in ieDf2 animals relative to wild type is that while the SC-CR may be important for RN formation in *C. elegans*, it is irrelevant to RN positioning, and other factors (i.e. the relative position of recombining partners) determine RN position.

Figure 6. The distribution of RNs in ieDf2 is clearly broader than in wild type, but to what extent can it be modeled as a wild-type distribution plus an axis-centered distribution? Perhaps a difference plot (i.e. ieDf2 RNs outside the wild type distribution) would be informative. Also, for display purposes, authors might want to consider summing the RN numbers for +0.2 and -0.2, +0.4 and -0.4, etc. and then plotting the values symmetrically around 0, since asymmetries in the plots are artifacts of the scan direction.

Final note. Figure 4 in Börner et al. shows the effect of zmm mutants on intersister joint molecules, and additional data are present in the literature.

Referee #1 Review

Report for Author:

Dr. Rog,

"Meiotic DNA exchanges are promoted by proximity to the synaptonemal complex"

The paper by Almanzar et al. describes the characterization of sister chromatid exchanges (SCEs) during meiosis in *C. elegans*. During meiosis, meiotic reciprocal exchanges called crossovers (COs), essential for chromosome segregation during meiosis I, are formed in tightly regulated ways to have COs between homologous chromosomes by a combined action of pro-crossover factors and meiotic chromosome structures. However, SCEs are much rare than interhomolog crossovers and hard to detect. Previously, the authors' group developed a method to analyze SCEs on meiotic chromosomes by differential labeling of EdU and

showed that meiotic SCEs are very low and elevated in some mutants (Almanzar et al. 2019). By using the novel cytological methods, the authors extended the previous study and showed that meiotic SCEs require pro-crossover factors such as ZHP-3. Moreover, interference for meiotic crossovers likely works on SCEs on chromatids. In addition, the authors performed a super-resolution microscopic analysis of the relative position of recombination nodules in the synaptonemal complexes (SCs). The experiments have been performed with great care and of good quality, and well-controlled. The most of results described are convincing. Compared to the authors' previous paper (Almanzar et al. 2019), this paper shows incremental progress in our understanding of the formation of SCEs in meiosis. Moreover, dependency of inter-sister exchange on the pro-crossover factors had been shown in budding yeast (the *zmm* mutants are defective in the formation of inter-sister joint molecules; Borner et al. 2004). In summary, without further data set to suggest a molecular mechanism on how SC-central regions regulate interhomolog and intersister recombination, the manuscript should go to more specialized journals.

Major points:

1. Please show CO (chiasma) frequencies in all mutants in the paper such as *syp-1K42E*, *zhp-3 (-)*, and *syp-1K42E zhp-3 (-)* mutants, and summarize all data sets of CO, SCE, and recombination nodules numbers in the mutant in a single table.
2. It is very important to demonstrate whether intersister recombination shows interference with the intersister and/or interhomolog recombination or not. The analysis in this paper did not fully address the question since the authors analyzed the recombination frequencies in a single nucleus, not on a single chromosome with reciprocal translocation (Figure 4). For interference between SCE-designated sites and CO-designated sites (or SCE vs SCE sites), it is very critical to show how many SCEs are generated on the non-homologous region of nT1/+ nuclei without the interference and then to analyze an actual number of recombination events such as recombination nodules in these translocated chromosomes. Otherwise, hard to evaluate the data set in Figures 4 and D.

Minor points:

1. Line 129-131: The word "coalescence" for CO interference is too speculative at this point. Please rephrase the reasoning for the subsequent experiments to reflect the nature of the experiment.
2. Line 153, Figure 2B: The *syp-2* shows 4.0 (in text) or 4.4 (in the Figure) SCEs. Which is the correct number?
3. Line 153, Figure 2E: Fig. 2E is not properly cited in the main text. This should be combined with Figure 1A.
4. Figure 2A and 2B: The data for *syp-3* and *rec-8* mutants are missing, which are described in the text (lines 137-143).
5. Figure 2C: Images of COSA-1 and HTIP-3 for *syp-3*, *rec-8*, and *syp-1K42E* mutants are missing. Please shows CR-staining in *syp1/2* and *syp-1K42E* mutants together with schematic figures.
6. Figure 5B: Rather than referring to the paper (Cahoon et al. 2019), it is important to show CR staining in the *rec-8* mutant under the same condition.
7. Line 251-252, the SC-CR directs the positioning of the recombination nodules: What data supports this claim?

Referee #2 Review

Report for Author:

Manuscript: "Meiotic DNA exchanges are promoted by proximity to the synaptonemal complex"

Overview: The manuscript by Almanzar et al. investigates the mechanism of how meiotic sister chromatids can promote exchanges in worms and find that the synaptonemal complex can regulate these exchanges by coalescing ZMM proteins, thereby building a recombination nodule required to promotes these exchanges. These findings suggests that the synaptonemal complex and recombination nodule can promote both homolog exchanges, as well as exchanges that can occur between sister chromatid when forced to do so. They show that the SC is the framework for laying down the pro-crossover factors whether that SC is between homologs, between arms of chromosome (in the fold-back of one chromosome) or assembled along sister chromatids. The data presented is mostly clear, but there are some comments below that should be considered and dealt with during a revision.

Points of consideration:

1. The distinction of SC-CR throughout the manuscript is a bit confusing, and care should be taken to make sure when/if SC-CR is used it is only referring to the central region of the SC, not the SC as a whole. For example, lines 64-67 states that homologs are aligned along their length by the central region of the synaptonemal complex, but aren't the homologs aligned along their length by the SC? Another example of this is on line 107 where it states that chromosome V does not associate with the SC-CR. Do you mean chromosome V doesn't associate with components of the central region, and therefore doesn't form an SC? It is assumed that the distinction is given because in some instances the axes, and perhaps the lateral elements, are there, but not the transverse filament components or the central element components. In these cases, it should be written as just the "central region" and not SC-CR. The words (central region) are usually not abbreviated.
2. While it is helpful to the reader to show the diagrams (like in Figure 1A) it is a bit confusing with the color choice, as the pink in Figure 1A represents something completely different than the magenta color in Figure 1B, although they are in the same location. One is SC and the other is the axis. As to not confuse the reader that the line down the middle of the homologs, as well as in other figures in the manuscript, all represent the same thing, it may be helpful to differentiate the colors more.

3. Line 100. Is 'homologous recombination' the correct term here if you are repairing off the sister chromatid? Perhaps a more correct way to word it would be "for repair of the double-strand break".
4. Figure 1A. for comparison purposes and to match the right side, a wildtype should be included in the bar graph.
5. Figure 1A legend. The authors mention analysis of *syp-1K42E* animals, but no data for that is shown in Figure 1A.
6. Figure 2A. The + should be changed to wildtype to be consistent with other figures.
7. Figure 2A-B. This reviewer would suggest you add *syp-2* data to A, add *syp-3(me42)* data to A and B, and leave out *zim-2* data in B.
8. Figure 2C. In the cartoon on the right, I believe you mean *syp-1* or *syp-2* and not *syp-1/syp-2*.
9. Lines 119-120. "mis-localize a mutated synaptonemal complex" should be clarified or explained more. Does the *rec-8* mutant result in a mutated SC? Or does the absence of *rec-8* lead to SC formation between sisters? Instead of mis-localize, it would be helpful to the reader to say it is between sisters, not homologs.
10. Line 136. Data for *ieDf2* should be 4.8 (per Figure 2A).
11. Line 137. Data for *syp-3(me42)* is missing from Figure 2A-B.
12. Line 139. For consistency and ease of reading, the number for RNs should come before SCEs, as it does for the two genotypes before it.
13. Lines 148-155. The data for *syp-2* seems to be lacking. No data is shown for *COSA-1* foci and no images are shown for this genotype in C-D. Full data should be shown for this genotype or left out completely.
14. Line 153. The SCE for *syp-2* should be 4.4 (per Figure 2B).
15. Throughout manuscript suggest changing "sister exchanges" to "sister chromatid exchanges" or the abbreviation of "SCE".
16. Figure 3B and text Line 163-165. Figure 3B is not the number of SCE/nucleus as it states in lines 163-165 but rather the % of single/double/no SCEs. The number of SCE/nuclei for *syp-1K42E* are found in Figure 2B and there is no data found for the *syp-1K42E zhp-3(-)* SCE/nucleus.
17. Figure legend 3C line 512 needs to reference both *syp-1K42E* and *syp-1K42E zhp-3(-)*, as Figure 3C is showing both.
18. Figure 4 legend. Line 525-there is no asterisk in figure 4B. Lines 527-528-there is not an interpretive diagram in figure 4C. The significant difference in Figure 4D has to be from the difference in the number of nuclei analyzed in the two genotypes. This reviewer agrees that this is not likely biologically significant, and therefore it is unfortunate that the genotypes were not equally analyzed. If this can be corrected it should be.
19. Figure S1 Line 564. It should be double-strand breaks or DSBs, not just simply breaks.
20. It would be helpful if there was a summary figure or model to show how the SC can promote the coalescence of RN component regardless of its location, which can then lead to COs or SCEs in certain instances. This should include the locations the RNs can be found as well.

Referee #3 Review

Report for Author:

Review for Almanzar et al.

During the mitotic cell cycle, most double strand break repair by homologous recombination uses the sister chromatid as a template, and noncrossovers are the favored product. During meiosis, use of the homologous chromosome (called the homolog) predominates, and crossover products are promoted, such that there is at least one crossover per homolog pair. The synaptonemal complex, a tripartite protein structure that forms between homologs, has been implicated both in promoting crossing over and recombination between homologs, and synaptonemal complex-associated protein ensembles, called recombination nodules, are thought to be the sites of crossing over. The current paper is an extension of Almanzar's previous *Current Biology* study, in which they developed a way to detect sister chromatic crossing-over in *C. elegans*, and showed that, in a case where one set of homologs remain unpaired, the synaptonemal complex localizes to sister chromatids in the unpaired region, and is associated with elevated sister chromatid exchange in those unpaired regions. In the current manuscript, Almanzar et al. extend this observation to different circumstances, including where all homologs are unpaired and form "foldback" synaptonemal, and extend the association between synaptonemal complex formation and crossing over between sister chromatids. They also present work that examines the question of whether crossover interference, which in *C. elegans* limits crossovers to one per homolog pair, extends to sister chromatid exchange, and also examines the spatial relationship between recombination nodules and the central element of the synaptonemal complex.

The scientific work in this manuscript is fundamentally sound, although correction may be needed in some circumstances, and the conclusions are to a large part supported by the data. The manuscript could be improved by placing it in a broader context, particularly in circumstances where meiotic recombination control in *C. elegans* appears to differ in degree from that in many other organisms. This absence of a broader context results in some text that is confusing and other text that is potentially misleading, because it often is not clear if authors are referring specifically to rules that operate specifically in *C. elegans* or those that are generalizable. For example, the general view of the paper that the synaptonemal complex recruits recombination nodule proteins ignores findings that, in other organisms, recombination nodules can assemble and function in the absence of SC. This absence of clear contextualization should be corrected in future versions.

The following comments are presented order of appearance in the manuscript; an asterisk (*) marks concerns of particular

importance.

*Line 1. The title over-generalizes, and does not accurately reflect the content of the paper. First, the title should make clear that this work examines events in *C. elegans*. Second, there are no data in the manuscript directly showing that crossovers at the DNA level occur near the synaptonemal complex; the data for this are circumstantial, and what is shown is that COSA-1 containing recombination nodules, which are taken as a proxy for crossovers, form in the proximity of the synaptonemal complex.

Introduction-general. While work in other organisms is mentioned in passing, the introduction is overly *C. elegans*-centric, especially for a manuscript with such a general title. For example, while in *C. elegans* synaptonemal complex forms in the absence of double strand breaks, and mutants lacking recombination nodule proteins still form synaptonemal complex, in many other organisms this is not true, and synaptonemal complex and recombination nodule formation are co-dependent.

Lines 63 and ff. A more detailed and nuanced description of the synaptonemal complex would help, including the point that the inter-axis part of the synaptonemal complex contains two substructures, the central element, and transverse filaments that connect the central element with the two chromosome axes. This is of particular importance, as recent work in budding yeast has shown that, in that organism, recombination nodule components are important for synaptonemal complex assembly, by recruiting central element proteins. Also, mutants lacking central element proteins and those lacking transverse filaments have different phenotypes with regards to crossing over.

Lines 70-77. This description is incomplete, and omits important examples. For example, in *Sordaria*, ZMM proteins form at allelic positions on separated axes at sites of "bridges" between aligned-at-distance homologs; in budding yeast transverse filaments, where synaptonemal complex does not form, ZMM proteins are localized to point associations between chromatid axes.

Lines 79-83. Should discuss work of Börner et al, who showed that crossover-specific inter-sister joint molecules are reduced or eliminated in mutants lacking recombination nodule proteins and in synaptonemal complex transverse filament mutants.

Results, general. It would help to mention frequencies of sister exchange in wild type cells as a reference.

Lines 111-114. References for this point, including data from yeast and mouse meiosis, would be useful.

Line 136. 4.7 nodules/nucleus vs 4.8 in Figure 2A.

Line 145. For this t-test, what is the comparison? Is it between *rec-8* and *him-8*?

Line 149. Worth naming the central element proteins (*syp-1* and *syp-2*).

*Lines 175-196. This analysis is problematic, as there is no mention of whether recombination nodules are in homologously- or heterologously-paired regions. Furthermore, in panel A of Figure 4, it is not clear whether the values represent chromosomes or nuclei; assuming that they are nuclei, the p value in the text is not correct for either comparisons to *him-8* (Fisher's exact test $p=0.11$) or to wild type ($p=0.29$). Moreover, unless there is a way to tell if a COSA-1 focus is in a region of homologous or heterologous synapsis, one can't tell if an interhomolog CO is preventing an inter-sister CO or vice versa. Is there some information missing here? Are the translocation chromosomes in *nT1/+* still recovered as chiasmata bivalents at diplotene/diakinesis? I would really help if the logic of this analysis were explained more clearly, making the assumptions and the logical threads of the argument more explicit. It currently is difficult to follow and not at all obvious what is meant.

*Lines 198-228. The STED analysis documents axis positions, not the location of DNA molecules, and neither the current analysis nor the previous analysis of Cahoon et al. show where the DNA molecules are, only where the axial element proteins are. In particular, it is not shown that sister chromatids are on one side of the SC in *ieDf2*. Absent a direct determination of where sister chromatids are, the arguments in this section need to be considerably moderated.

Line 331-333. My recollection is that the spatial relationship between recombination nodules and synaptonemal complex has been examined in organisms other than *Drosophila*, in particular plants by Stack/Anderson and Heyting. As was stated above, there are other documented situations where nodules form prior to synaptonemal complex or in its absence, and these need to be discussed. Finally, the last phrase leaves the impression that polycomplexes resemble recombination nodules-needs to be reworded.

*Lines 344-356. The argument in this paragraph ignores, as was mentioned above, prior data from fungi (budding yeast and *Sordaria*) that recombination nodules can form before tripartite SC is assembled and in mutants that never assemble SC. Thus, while a picture of ZMM protein coalescence in the context of the SC may be an attractive one for *C. elegans*, where the SC is present prior to and in the absence of events at the DNA level, the picture conveyed in this paragraph cannot be generalized and is arguably misleading for other organisms. Furthermore, the description of the SC-CR as a phase-separated compartment in itself is a misuse of terminology, because phase separation formally describes the dynamics of a multi-component system where

at least one component is excluded from assemblies of the other(s) (thus the word *separation*), and while the principle of coalescence has been demonstrated for the SC, exclusion has not been. Correcting this terminology is essential, and moreover will not markedly affect the actual conclusions of the paper, which can all be understood in terms of association and coalescence.

January 9, 2023

Re: Life Science Alliance manuscript #LSA-2023-01906-T

Dr. Ofer Rog
University of Utah
School of Biological Sciences and Center for Cell and Genome Sciences
257 South 1400 East
Salt Lake City, UT 84112

Dear Dr. Rog,

Thank you for submitting your manuscript entitled "Meiotic DNA exchanges in *C. elegans* are promoted by proximity to the synaptonemal complex" to Life Science Alliance. We invite you to submit a revised manuscript addressing Reviewer 2 and 3's remaining comments.

Thank you for this interesting contribution to Life Science Alliance. We are looking forward to receiving your revised manuscript.

Sincerely,

B. MANUSCRIPT ORGANIZATION AND FORMATTING:

Referee #1:

Dr. Rog,

"Meiotic DNA exchanges in *C. elegans* are promoted by proximity to the synaptonemal complex".

This is a re-review of the revised manuscript for a previous submission by the authors

This paper by Almanzar shows a connection of chromosome structure compartment with sister chromatid (SCE) in *C. elegans* meiosis. The authors analyzed SCE formation in various mutants and found that SCE formation relies on the assembly of components of the central region of the synaptonemal complex (SC), a meiosis-specific ladder-like chromosome structure that juxtaposes two homologous axes, which would provide a nuclear compartment (domain) for the assembly of recombination nodules (RNs) containing pro-crossover proteins. The experiments were carried out with good quality and most of the results described are solid.

However, in terms of our understanding of the molecular mechanism of meiotic recombination including SCE, the results (and the proposal) are almost experimental confirmation of the previous ideas. Among them, the following are new. First, the authors found the RN in the region where SCE occurs (1-to-1 correspondence). Second, inter-homolog recombination would communicate with inter-sister recombination, manifested by interference. The latter idea could be extended by studying the super-resolution structure of synapsed translocated chromosomes in nT1/+ strain (Figure 4B)-how SC central regions are assembled on the chromosomes such as homologous synapsed region and non-homologous synapsed regions. Alternatively, as shown in *Arabidopsis thaliana* (Durand et al. Nature Commun, 2022), in respect of coarsening in RNs in the SC compartment, it would be interesting to see the gene dosage effect of the pro-crossover factor such as the overexpression on the distribution/number of RNs and/or SCEs along chromosomes, although this is a bit out of the topic in the paper.

In conclusion, the revision is not enough to change the previous conclusion by this reviewer that, without new data sets, this paper shows incremental progress in our understanding of the molecular link of meiotic recombination to the chromosome structure/compartment.

Referee #2:

Revision assessment: The authors were able to address most of the concerns from the first set of reviews, making this revision now suitable for publication. In

addition to clarifying some of the figures, the authors have provided additional experiments to address concerns by the first set of reviews (addressing interference, conformation of the sisters in *ieDf2*, and repair intermediate localization in regard to the central region). Furthermore, they have now included appropriate citations and work from other model systems and discuss similarities and differences between model organisms. The only concern to be raised is perhaps in the wording of the conclusion that the SC regulates SCEs. While the SC can promote SCEs when built between sisters, the SC would normally not regulate their occurrence but block those exchanges from ever occurring since it forms between homologs. In addition, these studies could be addressing the processes that prevent SC from forming between sisters and not the regulation per se of SCEs.

We edited the conclusions to clarify this point. Specifically, we changed the abstract to highlight that the SC *can* regulate SCEs. We use a similar formulation in the results (line 91) and the discussion (e.g., line 327).

Referee #3:

In their revision, Almanzar et al. have addressed many of the concerns expressed in previous reviews, and the manuscript has been improved accordingly. The manuscript remains somewhat bifurcated, with the first part containing an exploration of factors that contribute to sister chromatid exchange (SCE) and formation of recombination nodules (RNs) involving sister chromatids. The second part examines the location of RNs relative to axes and synaptonemal complex central elements in either homologously or non-homologously synapsed chromatids. While both parts of the manuscript contain worthwhile contributions to the literature, neither alone or in combination provide conclusive support to the conclusion regarding the relationship between crossovers and the synaptonemal complex stated in the title or abstract.

Comments (referral is to line numbers):
61-63. This sentence is incomplete.

Fixed.

122 and ff. Why so much discussion of *zim-2* and *him-8* when SCE levels in the two are not statistically different?

We clarified the text to indicate the lack of statistical significance, as well as possible reasons for the small effect. Since potential difference between chromosomes are of interest to the worm meiosis community, we are opting to keep this data in the manuscript.

174 and ff. ZHP-3 depletion in *syp-1K42E* animals results in a 2.5-fold reduction in SCE, but a complete loss of RNs. However, unlike in other SCE-promoting mutants/variants, where SCEs and RNs are in roughly equal numbers, *syp-1K42E* mutants display a 2-fold excess of RNs over SCEs, suggesting that the two are already uncoupled in this mutant. It would also be useful to compare the 2.5-fold reduction in SCE in auxin-depleted ZHP-3 *syp-1K42E* mutants with the fraction of interhomolog crossovers (i.e. bivalents at diplotene/diakinesis) remaining upon auxin-depletion of otherwise wild-type cells either.

This is indeed an interesting point, which we have now addressed. In the case of inter-homolog crossovers in AHP-3-depleted animals - these are completely lacking and no chiasmata are observed.

212 and ff. SCEs in *nT1/+* animals are not significantly different from those in WT, *him-8* or *zim-2*, but the statistical power to detect differences is not stated. This is important for the assessment of to what extent an interhomolog crossover could possibly affect SCE.

We have clarified that important caveat in the text.

334 and ff. While Woglar & Villeneuve may have suggested a stereotypical RN positioning relative to the SC-CR, previous cytological data have suggested a broader distribution.

That is true, and we apologize for not clarifying that this sentence refers specifically to worms. We have added those reference and also discuss the cytological observations from other organisms below in the Discussion (now lines 409-411). Also see lines 249-253.

415 and ff. An alternative explanation for the broader distribution of RNs in *ieDf2* animals relative to wild type is that while the SC-CR may be important for RN formation in *C. elegans*, it is irrelevant to RN positioning, and other factors (i.e. the relative position of recombining partners) determine RN position.

We agree with this point, and have alluded to it in the conclusion (beginning now in line 430). We have now further clarified this point.

Figure 6. The distribution of RNs in *ieDf2* is clearly broader than in wild type, but to what extent can it be modeled as a wild-type distribution plus an axis-centered distribution? Perhaps a difference plot (i.e. *ieDf2* RNs outside the wild type distribution) would be informative. Also, for display purposes, authors might want to consider summing the RN numbers for +0.2 and -0.2, +0.4 and -0.4, etc. and

then plotting the values symmetrically around 0, since asymmetries in the plots are artifacts of the scan direction.

This is an interesting point which we agree with. When carrying out our analysis, we have attempted to distinguish between the fits of multiple models of RN distribution vis-a-vis the SC-CR. Unfortunately, we did not feel our data has sufficient power to robustly distinguish between different models, so we have decided to leave this analysis out of the current manuscript. We have now added text to clarify these limitations.

Final note. Figure 4 in Börner et al. shows the effect of *zmm* mutants on intersister joint molecules, and additional data are present in the literature.

Börner et al. indeed measures inter-sister double-Holiday junctions in various *zmm* mutants. However, the very slight differences are not quantified and, judging by their appearance in the published data, are unlikely to be different or significant.

January 10, 2023

RE: Life Science Alliance Manuscript #LSA-2023-01906-TR

Dr. Ofer Rog
University of Utah
School of Biological Sciences and Center for Cell and Genome Sciences
257 South 1400 East
Salt Lake City, UT 84112

Dear Dr. Rog,

Thank you for submitting your revised manuscript entitled "Meiotic DNA exchanges in *C. elegans* are promoted by proximity to the synaptonemal complex". We would be happy to publish your paper in Life Science Alliance pending final revisions necessary to meet our formatting guidelines.

-please add a separate Conflict of interest statement to your main manuscript text

Figure Check:

-please add scale bars to Figure S3 A and B

LSA encourages authors to provide a 30-60 second video where the study is briefly explained. We will use these videos on social media to promote the published paper and the presenting author (for examples, see <https://twitter.com/LSAjournal/timelines/1437405065917124608>). Corresponding or first-authors are welcome to submit the video. Please submit only one video per manuscript. The video can be emailed to contact@life-science-alliance.org

A. FINAL FILES:

B. MANUSCRIPT ORGANIZATION AND FORMATTING:

Sincerely,

January 11, 2023

RE: Life Science Alliance Manuscript #LSA-2023-01906-TRR

Dr. Ofer Rog
University of Utah
School of Biological Sciences and Center for Cell and Genome Sciences
257 South 1400 East
Salt Lake City, UT 84112

Dear Dr. Rog,

Thank you for submitting your Research Article entitled "Meiotic DNA exchanges in *C. elegans* are promoted by proximity to the synaptonemal complex". It is a pleasure to let you know that your manuscript is now accepted for publication in Life Science Alliance. Congratulations on this interesting work.

DISTRIBUTION OF MATERIALS:

Again, congratulations on a very nice paper. I hope you found the review process to be constructive and are pleased with how the manuscript was handled editorially. We look forward to future exciting submissions from your lab.

Sincerely,
